# Long-Term Efficacy and Safety of Dupilumab in Patients with Atopic Dermatitis: A Single-Centre Retrospective Study

**Michela Ortoncelli †, Nicole Macagno †, Luca Mastorino * , Federica Gelato, Irene Richiardi, Giovanni Cavaliere, Pietro Quaglino and Simone Ribero**

Dermatology Clinic, Medical Sciences Department, University of Turin, 10121 Torino, Italy; mortoncelli@cittadellasalute.to.it (M.O.); nicole.macagno@edu.unito.it (N.M.); federica.gelato@gmail.com (F.G.); irene.richiardi@edu.unito.it (I.R.); gcavaliere@cittadellasalute.to.it (G.C.); pietro.quaglino@unito.it (P.Q.); simone.ribero@unito.it (S.R.)

* Correspondence: lucamastorino02@gmail.com
† These authors contributed equally to this work.

**Abstract:** Introduction: There are few long-term effectiveness and safety data for dupilumab in the treatment of atopic dermatitis (AD). The aim of this study was to evaluate efficacy and safety of dupilumab for up to three years after treatment initiation. Materials and Methods: We collected data from patients $\geq$ 12 years with severe AD who started dupilumab at the Dermatology Clinic of the Turin University Hospital between December 2018 and October 2022. Clinic and patient reported outcomes were evaluated from baseline, up to 3 years (T9), every 4 months. Results: A total of 418 patients were observed. A progressive decrease in the meanEASI was observed: from 23.64 at baseline to 2.31 at T9. Similar trends were observed in patients' reported outcomes. The achievement of EASI75 and EASI90 was observed in 75.58% of patients and 53.49%, respectively, at T1 (4 months), and in 92.55% and 80.85% at T9; DLQI 0/1 was achieved at T9 in 61.7%. Mean NRSpp $\leq$ 4 was achieved at T9 in 91.5% (86 out of 94 patients). The most common adverse event was conjunctivitis occurring in 13% of patients on average at each timepoint analyzed. Conclusions: Dupilumab proved to be effective and safe for the treatment of AD in clinical practice, up to 3 years.

**Keywords:** atopic dermatitis; dupilumab; EASI; DLQI; itch





## 1. Introduction

Atopic dermatitis (AD) is one of the most common inflammatory skin diseases. This condition is characterized by a chronic and relapsing course with flareups of itchy eczematous lesions on dry skin. The prevalence of AD has increased in most continents and has reached a stable plateau in Europe and North America with prevalence in children reaching 15–20% and in adults 1–3% [1]. The hallmark of AD is itching, which is the main symptom and can reduce quality of life [2,3].

The pathophysiology of atopic dermatitis is complex and multifactorial and includes genetic disorders, epidermal barrier defects, alterations in the immune response, and environmental alterations (such as disruption of the skin's microbial balance) that cause skin barrier abnormalities and immune dysfunction that are considered crucial to the pathogenesis of AD [4,5].

Two main theories have been proposed to try to explain the cause of AD: the inside-out hypothesis and the outside-in hypothesis.

The inside-out hypothesis proposes that it is the allergic trigger that leads to a weakening of the skin barrier that facilitates the introduction and presentation of allergens. This would suggest that inflammation is responsible for a compromised skin barrier, which leads to increased penetration of allergens and microbes that cause the lesion.

The outside-in hypothesis assumes instead that there is an intrinsic defect in the epithelial cells, i.e., a defect in the epidermal barrier, due to which allergens or irritants

can easily penetrate the epidermal barrier and secondarily induce an immunological reaction [6]. For example, the downregulation of filaggrin (FLG), which is necessary for the proper functioning of the skin barrier, could make the skin more susceptible to immune dysregulation and lead to AD.

Since the two theories are unlikely to be exclusive, the more truthful situation is that both play a role in the development of the disease [7,8].

The diagnosis of atopic dermatitis is essentially clinical. A thorough personal and family clinical history is essential, assessing the morphology and location of the lesions and taking into account accompanying clinical signs.

Current commonly used criteria include dermatitis associated with pruritus, which must be associated with at least three of the five major criteria below:

- Positive personal history of dermatitis localized to the antecubital folds, popliteal cords, ankles, periocular area, and neck.
- Positive personal history of asthma and/or allergic rhinitis (or, if the child is less than four years old, positive family history of atopy with an affected first-degree relative).
- Positive personal history of skin xerosis in the last period.
- Injuries to the flexural folds or, in children up to the age of four, to the cheeks, forehead, and limbs.
- Onset of symptoms before the age of two, only for subjects from the age of four years [9,10].

AD is characterized by itching, erythema, induration, and scales, but these features are also typical of several other conditions that can mimic, coexist with, or complicate AD. These include inflammatory skin conditions, infections, infestations, neoplasms, genetic disorders, immunodeficiency disorders, nutritional disorders, graft-versus-host disease, and drug eruptions. Knowledge of the spectrum of these diseases and their distinguishing features is crucial for correct and timely diagnosis and optimal treatment [11].

Several treatments are currently available, and their use depends on disease severity. For all stages, even in disease-free intervals, the use of emollients is recommended, not using soap, and avoiding triggers. In cases of mild-to-moderate AD, the treatment is based on topical calcineurin inhibitors or topical corticosteroids. An adjunct therapy can be phototherapy, preferably with UVB 311 nm or UVA1. In severe AD requiring systemic treatment, systemic corticosteroid is used in the management of flareups, but the only traditional on-label long term treatment is cyclosporine, ranging from 3 to 6 mg/kg/die [12,13]. Methotrexate and azathioprine are second-line treatments to be used if recommended first-line treatments are not available. A number of modern treatment options have recently become available, including the biological drugs tralokinumab and dupilumab, currently available and recommended for the treatment of severe atopic dermatitis, and the Janus kinase inhibitors abrocitinib, baricitinib, and upadacitinib. Among biological drugs, dupilumab was the first to be introduced in the treatment of adults and later children > 6 years old with moderate–severe AD when systemic therapy is needed. Dupilumab, a recombinant human IgG4 monoclonal antibody that inhibits interleukin (IL)-4 and IL-13 signal transduction, is indicated for the treatment of moderate-to-severe AD in adults and adolescents 12 years of age and older. It inhibits IL-4 signal transduction through the type I receptor (IL-4R$\alpha$/$\gamma$c), and both IL-4 and IL-13 signal transduction through the type II receptor (IL-4R$\alpha$/IL-13R$\alpha$). IL-4 and IL-13 are key factors in the pathogenesis of type 2 inflammatory diseases such as atopic dermatitis, asthma, and chronic rhinosinusitis with nasal polyposis (CRSwNP). Blocking the IL-4/IL-13 pathway with dupilumab in patients reduces many of the mediators of type 2 inflammation [14].

Data in the literature demonstrate efficacy, reducing signs and symptoms of the disease and improving quality of life in placebo-controlled phase III trials, clinical trials, real life studies (REF), and safety in patients with moderate–severe AD treated with dupilumab as continuous treatment [15–20].

Many studies have claimed that dupilumab is effective and safe in the short term, but few studies have demonstrated the maintenance of efficacy and safety of this drug even in the long term by showing efficacy and safety data about a year or so later [15–17].

The reduction in EASI in treatment with dupilumab appears to be correlated in some studies with the reduction in pruritus and DLQI, although other parameters related to quality of life and psychological impact such as the HADS depression and anxiety test and the SF-36 test did not show similarly strong correlations. These data emphasize the high psychological impairment brought about by the awareness of the chronicity of the disease, and the need for an equally chronic treatment, not to mention the feelings of anxiety often associated with injection therapy. The values of immunoglobulin-E, lactate dehydrogenase (LDH), and transepidermal water loss (TEWL) also change during the course of dupilumab therapy, but were not evaluated in our study [21,22].

The aim of this study is to evaluate the efficacy and safety of dupilumab in the continuous treatment of moderate–severe AD for up to three years after initiation of treatment. In addition, several scores that also assess subjective response to dupilumab were evaluated throughout the treatment period with scores assessing quality of life, itching, and sleep disturbance.

## 2. Materials and Methods

In this retrospective study, data were prospectively collected from patients aged $\geq$ 12 years with moderate-to-severe AD who started treatment with dupilumab at the Dermatology Clinic of the Turin University Hospital between December 2018 and October 2022. All the patients were followed up until February 2023.

Adult patients had to have an Eczema Area Severity Index (EASI) $\geq$ 24, while adolescents required EASI $\geq$ 24 or less with the presence of one of the following criteria: localization in sensitive or visible areas, Numerical Rating Scale peak of pruritus (NRSpp) $\geq$ 7, Dermatology Life Quality Index (DLQI) $\geq$ 10.

Dupilumab was administered with standard dosing (600 mg as starting dose, followed by 300 mg every 14 days).

Several parameters were evaluated: mean EASI score at baseline, the achievement of 75% improvement in the Eczema Area and Severity Index (EASI 75), the achievement of 90% improvement in the Eczema Area and Severity Index (EASI 90), mean NRSpp, mean sleep disturbance Numerical Rating Scale (NRSsd), mean patient-oriented eczema measure (POEM), mean DLQI at baseline (T0), at 16 weeks (T1), at 32 weeks (T2), at one year (T3), at two years (T6), and at three years (T9).

For this analysis, epidemiological data (demographic and disease characteristics and medical history), disease severity (EASI, NRSpp, NRSds), DLQI, POEM, history of cancer and previous treatments were summarized using descriptive statistics. Descriptive statistics were used to evaluate the dataset according to the number of patients and their percentage proportion in the groups related to the categorical variables; mean and standard deviation (SD) were used for continuous variables.

The present study was approved by our Institutional Review Board under the protocol named SS-DERMO-13.

## 3. Results

### 3.1. Patient Characteristics at Baseline

At baseline (T0), data were collected from 418 patients including 36 adolescents, 382 (31 adolescents) at 4 months after the start of treatment, 336 (21 adolescents) at 8 months (T2), 297 (16 adolescents) at 12 months (T3), 268 (13 adolescents) at 16 months (T4), 226 (11 adolescents) at 20 months (T5), 182 (1 adolescent) at 24 months (T6), 146 (0 adolescents) at 28 months (T7), 124 (0 adolescents) at 32 months (T8), 92 (0 adolescents) at 36 months (T9).

Among patients included in the study, 226 (53.94%) were males, the mean age was 39.2 (sd $\pm$ 17.43) years old, and the mean age of onset of AD was 13.5 (sd $\pm$ 20.6) years

old. Of the 418 patients, 271 (65.9%) had childhood onset, 149 (37.9%) had family history of atopy, 25 (8.8%) had a manifestation of prurigo excoriata, 96 (23%) had history of allergic conjunctivitis, 79 (19.2%) had had recurrent herpetic infections, 8 (1.9%) had positive history of parasitic infections, and 3 (0.73%) had diagnosed ichthyosis (Table 1).

**Table 1.** Baseline and demographic characteristics of the population.

| Demographic Characteristics | |
|---|---|
| | **N°/%** |
| Sex (M) | 226 (53.94%) |
| Age | 39.2 (sd 17.43) |
| Age of onset | 13.5 (sd 20.6) |
| Childhood onset | 271 (65.9%) |
| Familiarity | 149 (37.9%) |
| Prurigo excoriate | 25 (8.8%) |
| Allergic conjunctivitis | 96 (23%) |
| Recurrent herpetic infections | 79 (18.9%) |
| Parasitic infections | 8 (1.9%) |

All patients performed topical steroid therapy, 46.6% of patients used topical immunomodulator therapy, 10.2% phototherapy, 96% systemic steroid therapy, 84% cyclosporine therapy, and 9% received omalizumab therapy.

*3.2. Clinical Activity*

As shown in Figure 1a, the mean EASI at baseline was 23.64 (ds $\pm$ 10.44), falling to 3.69 (ds $\pm$ 4.95) at T1; the downward trend persisted until T9 with a mean EASI of 1.4 (sd $\pm$ 2.63). The mean follow-up time was 24.9 months (ds $\pm$ 13.9), while the median follow-up time was 24 months (quartile range 12–38).

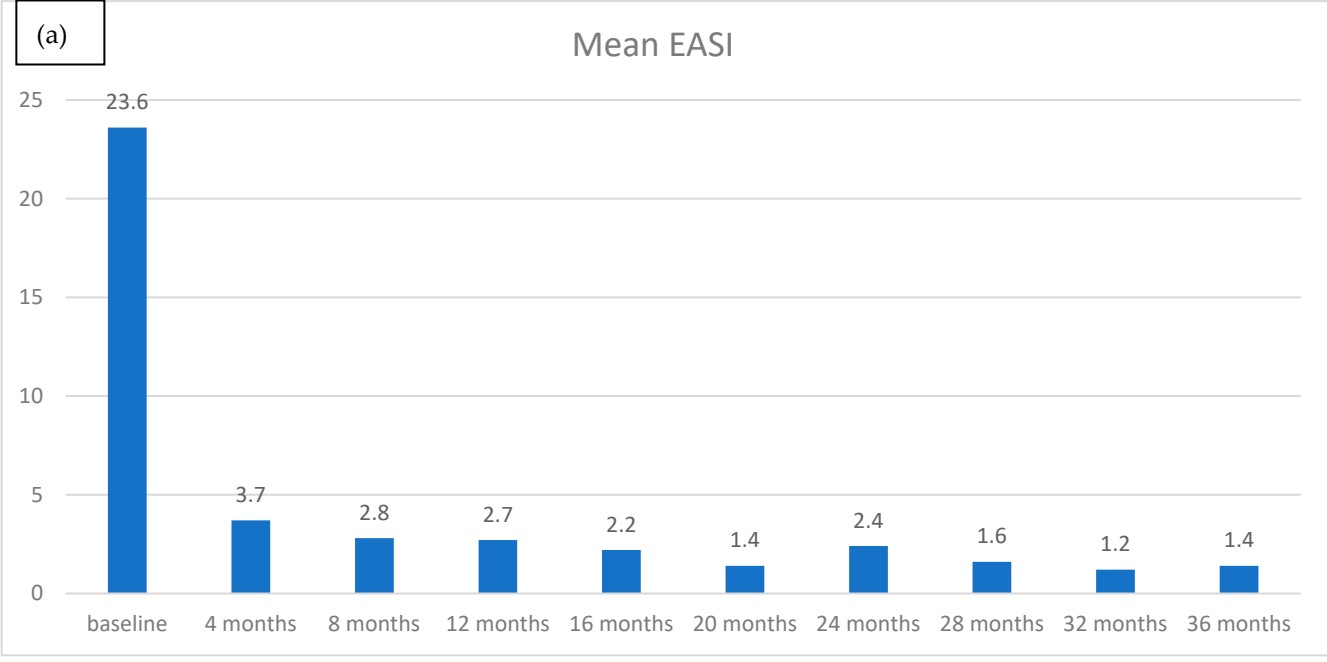

**Figure 1.** *Cont.*

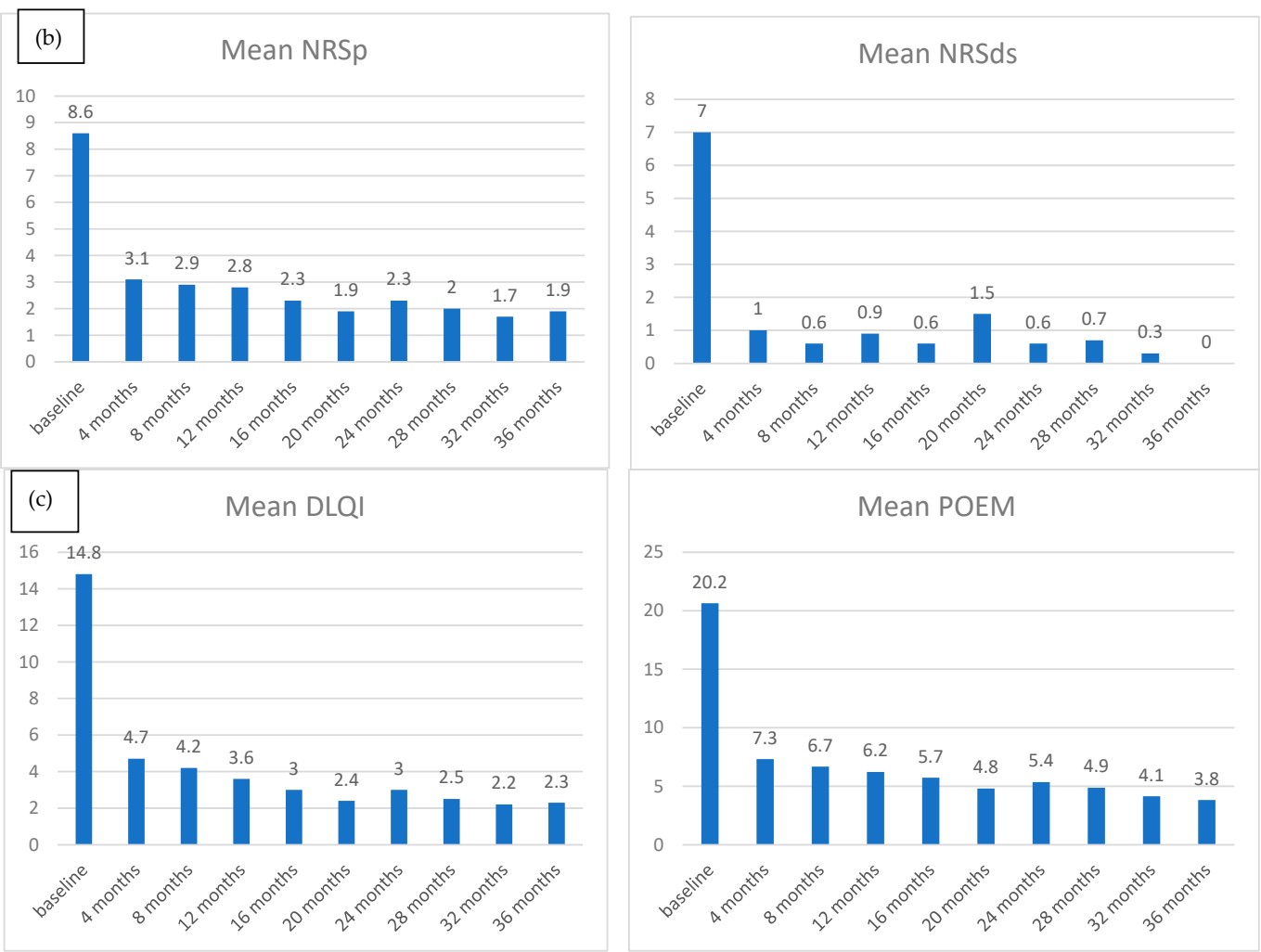

**Figure 1.** (**a**) Mean EASI reduction from baseline to T9; (**b**) mean NRSp and NRSd reduction from baseline to T9; (**c**) mean POEM and DLQI reduction from baseline to T9.

At T1 260 patients out of 344 (75.58%) achieved EASI75, at T2 80.26%, at T3 80.23%, at T 88.83%, and at T9 92.55%. Regarding EASI90, the trend is similar to EASI90 achieved at T1 in 53.49% of patients, at T2 in 53.72%, at T3 in 56.2%, at T6 in 74.86%, and at T9 in 80.85% (Figure 2a).

Similar to EASI, patient-reported outcomes showed a general trend of decreasing mean value during the months of treatment.

Regarding the peak pruritus numerical rating scale (NRSpp), at T0, the mean value was 8.56 (ds ± 2.32), reduced at 3.08 (ds ± 2.53) at T1 and 1.89 (ds ± 2.09) at T9 (Figure 1b). Mean NRSpp ≤ 4 was achieved at T1 in 71.3% of patients, at T2 in 75.0%, at T3 in 76.61%, at T6 in 82.7%, and T9 in 91.5% (Figure 2b).

As for quality of life, a similar decreasing trend was found: mean DLQI was 14.83 (ds ± 7.16) at T0, falling to 4.71 (ds ± 4.96) at T1 and achieving a mean value of 2.31 (ds ± 3.18) at T9 (Figure 1c). DLQI 0/1 was achieved at T1 in 68.06% of patients (228 of 335 patients), at 32 T2 in 59.73%, at T3 in 55.42%, at T6 in 50.56%, and at T9 in 61.7% (Figure 2b).

At T0, the mean POEM value was 20.64 (ds ± 6.1), at T1 7.31 (ds ± 5.79), and at T9 3.82 (ds ± 5.17) (Figure 1c).

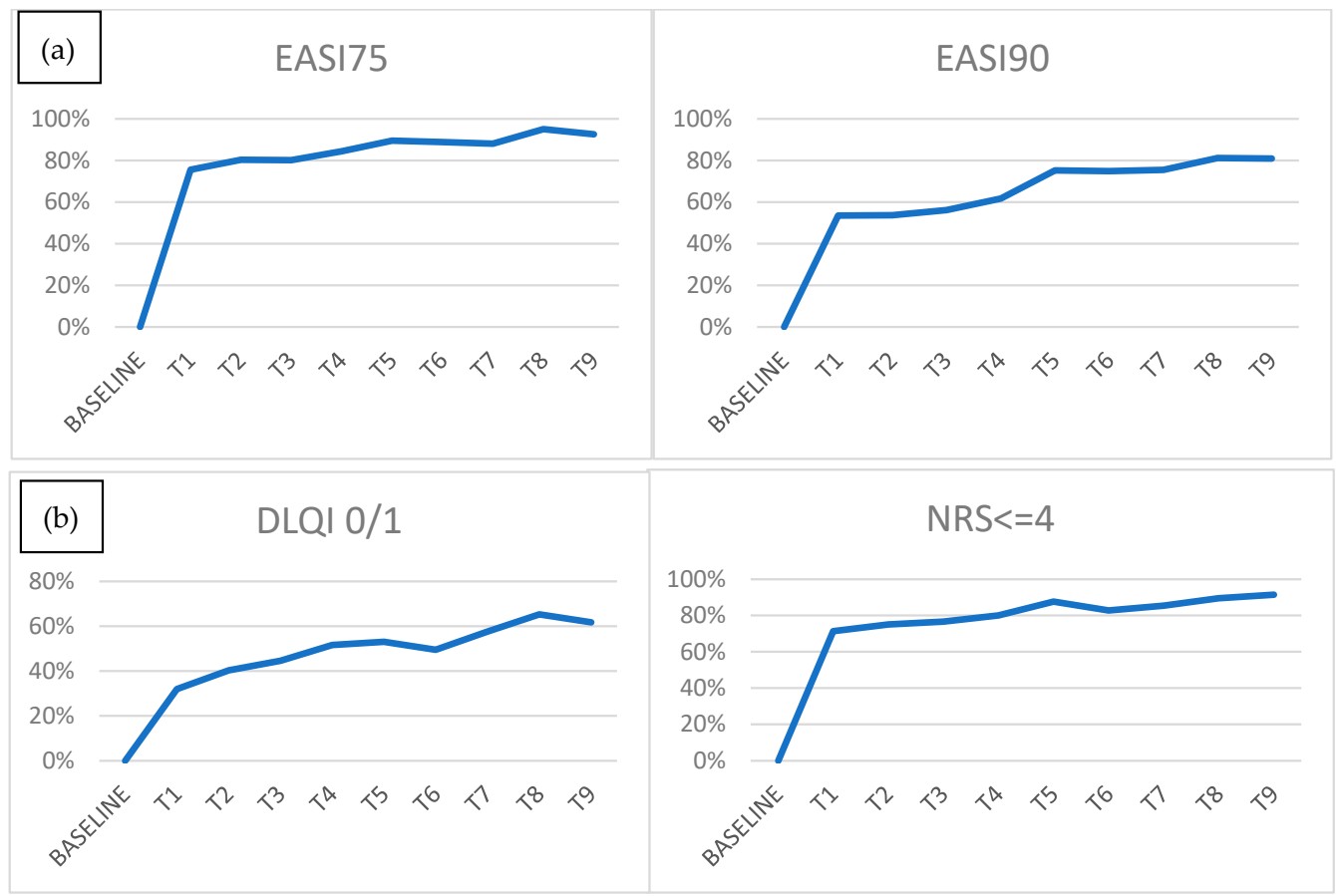

**Figure 2.** (**a**) EASI75 and 90 achievements at each time point; (**b**) DLQI 0/1 and NRSp <= 4 achievements at each time point.

The trend of some laboratory values including total immunoglobulin E (IgE) level and absolute value of eosinophils was also analyzed. A subsequent general downward trend was described for these values. At T0, the mean IgE was 3239.24 (ds ± 5020.4), at T1 1929.98 (ds ± 3138.4), and at T9 209.3 (ds ± 308.45).

At T0 the mean eosinophils was 2.1 (ds ± 19.63), at T1 0.64 (ds ± 0.81), and at T9 0.38 (ds ± 0.24).

At baseline, the mean LDH value was 320.88 (ds ± 144.2), at T1 236.27 (ds ± 96.03), and at T9 198.29 (ds ± 61.55) (Figure 3).

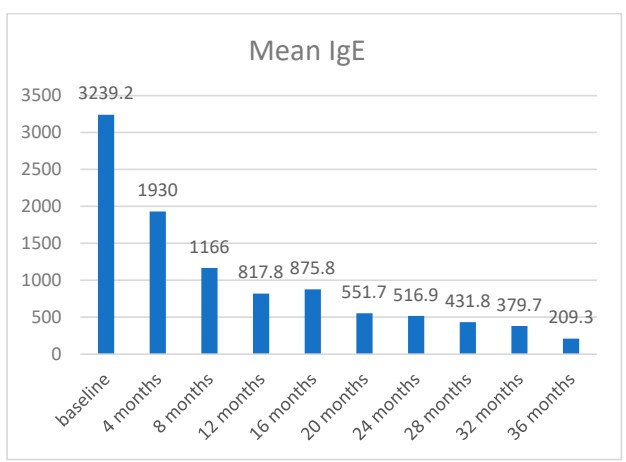

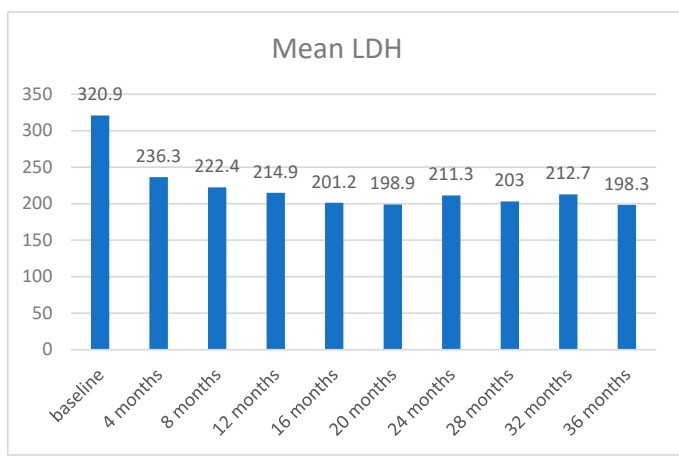

**Figure 3.** Mean LDH and mean total IgE at each time point.

*3.3. Safety*

The most common adverse event was conjunctivitis, which occurred in 5% (11 patients out of 210) one month after the start of treatment, in 13% (45 patients out of 355) at 4 months, in 12% (39 patients out of 330) at 8 months, in 12% (39 patients out of 333) at 12 months, in 6% (12 patients out of 214) at 2 years, and in 8% (9 patients out of 120) at 3 years after the start of treatment with dupilumab.

The prevalence of conjunctivitis was 14.10% of patients at T1, 14.9% of patients at T2, 13.5% at T3, 11.0% at T6, and 13.0% at T9. The second most frequent adverse event was recurrent herpetic infections, with a prevalence of 5.5% of patients at T1, 6.5% at T2, 5.7% at T3, 7.7% at T6, and 7.6% at T9.

Other less frequent adverse events were headache, alopecia, typical red face, myalgias, and arthralgias and dizziness (Figure 4).

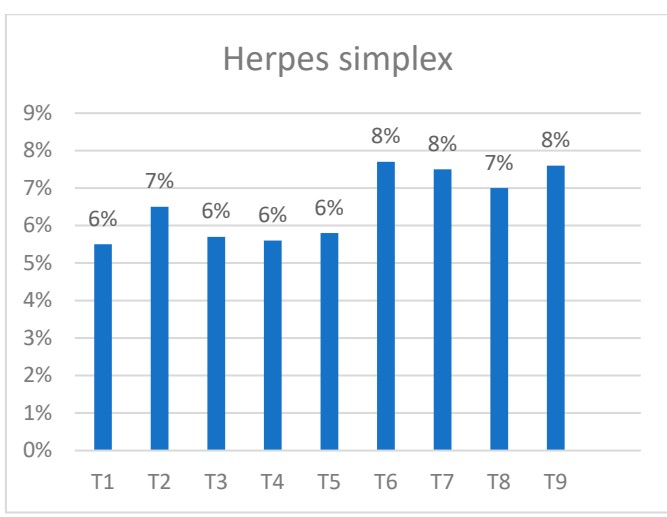 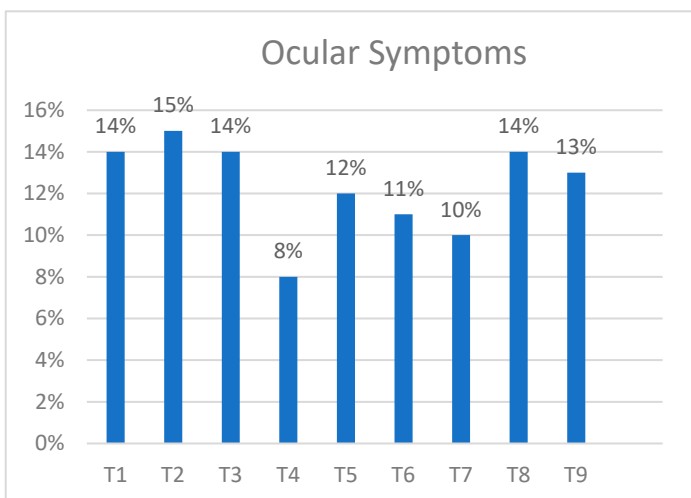

**Figure 4.** Percentage of ocular symptoms and herpes infections reported by the patients at each time point.

## 4. Discussion

The new findings from this study concern the safety profile of dupilumab and the maintenance of its efficacy even over a long period of time. There are several studies in the literature demonstrating such data 52–76 weeks after the start of treatment, but few studies demonstrate its maintenance in the long term [15,16,20,23]. The strength of our study is its evaluation of the safety and efficacy of dupilumab with data up to three years after treatment initiation on a large cohort of 418 patients.

The results obtained demonstrate a good safety and efficacy profile in moderate-to-severe AD even with prolonged treatment. In our case series, the majority of patients improved, showing how dupilumab can be a valid treatment option, leading to objective improvements in disease severity that were measured using the EASI score, but also subjective improvements in pruritus and quality of life measured using different scores (NRSpp, NRSds, POEM, DLQI).

Therefore, the strengths of this study are the large sample size and the long follow-up period that allowed for results based on continuous treatment with dupilumab for 3 years.

Regarding the efficacy of dupilumab, the results show good control of signs and symptoms of the disease with progressive reduction in mean EASI and NRSpp scores up to three years of treatment, with achievement at three years of EASI75 in 92.55% of patients and EASI90 in 80.85%. They also demonstrated an improvement in quality of life with achievement of DLQI 0/1 in 61.7% of patients at three years.

Overall, the reduction in the mean values analyzed reveals a trend towards a rapid therapeutic response as early as 4 months after the start of treatment with substantial

plateauing, especially with regard to EASI. The patient-reported outcomes, although revealing a trend towards plateauing, seem to show a possible further reduction beyond the time we analyzed.

The trend in relative indices appears slightly progressive after an initial major response, with variable peaks for EASI 75 and 90, NRSp <= 4 and quality-of-life tests. This finding is in line with the recurrent nature of the disease, with flares that are difficult to predict on the basis of stress and seasonality, and that although reducing with dupilumab treatment, they tend to persist sporadically.

In our case series, mean IgE shows a progressive reduction even beyond the first year of treatment, with mean values below the upper limit of normal not yet reached at 36 months. In any case, the figure reveals a relevant efficacy even in a population suffering from so-called extrinsic AD in which the impact of the exposome is significant and could compromise the response.

The LDH trend after an initial sharp reduction shows substantial stabilization at normal levels, and in a previous study of ours, it was noted that this inflammatory index was not a good proxy for the other parameters, and its periodic dosage was recommended against, adding no useful information for therapeutic management.

In our study, we referred to eye disorders as 'conjunctivitis'; within this umbrella term we include the so-called DIOSD, i.e., the surface disorder related to dupilumab treatment. It is well known that dupilumab inhibiting OX40-L and part of the protein present in the tear film can induce a surface disorder of the eye or aggravate a previous allergic conjunctivitis. In a recent study, however, no substantial difference from healthy controls was observed in atopic patients treated with dupilumab, nor a progressive worsening in the surface condition [24].

In a previous study, we described similar effectiveness of dupilumab up to 36 months irrespective of comorbidities whether allergic or not [25].

These are real-life data that support data obtained by Beck et al. in the multicenter clinical trial, who had a mean EASI at T0 of 33.4 and 1.5 at week 148 compared to the mean EASI obtained in our real-life data of 23.64 at T0 and 1.4 at T9. Likewise, the mean NRSpp of 7.2 at T0 and 2.1 at 148 weeks is comparable with our mean NRSpp of 8.56 at T0 and 1.89 at 156 weeks [26].

Regarding the trend of blood values of eosinophils, the level of IgE and LDH, for all three blood values, we see a progressive decrease in value from baseline to three years later in our case series. The study published by Zhou et al. shows that the level of eosinophils, LDH, and IgE increased temporarily in the first 4 weeks, then decreased and stabilized during dupilumab treatment. They also argued that there is no relationship between the levels of eosinophils, IgE, LDH, and the therapeutic efficacy of dupilumab [27].

A recent study by Kimball et al., RELIEVE-AD showed a marked reduction of over 50% in topical medication in atopic patients treated with dupilumab contacted at 36 months after initiation, with marked improvement in pruritus and skin manifestations, with an overall satisfaction rate of close to 75%. Patients reported a rapid response in the first year of treatment and subsequent stabilization of disease, as in our case series [28].

Bosma et al., in the Dutch TREAT NL study, reported a progressive reduction in EASI, POEM, and DLQI up to 48 weeks, with values that were curiously even six mean points higher than ours at the same endpoints for EASI and four to two mean points for the quality of life tests. These observations may raise concerns against evaluative homogeneity between various centers and countries in the description of the disease, and different psychological impacts between populations living in different geographical regions even though on the same continent [29].

The data are consistent with the known safety profile of dupilumab [26,30,31]. A typical side effect is conjunctivitis, which has already been described in the literature. Most conjunctivitis events reported in this study were mild-to-moderate in severity and did not result in treatment discontinuation [32].

In our study, conjunctivitis affected an average of about 13% of patients at each timepoint analyzed. Similarly, recurrent herpes infections affected an average of 6.5% of patients at each timepoint analyzed. In both cases, these data do not seem to identify a trend of increasing frequency with increasing time of months of treatment with dupilumab. In line with our data, a two recent meta-analysis of real-life studies also reported a similar incidence rate of conjunctivitis (26.1% and 8.0%, respectively) and a herpes incidence rate of 5.8% and 6.1%, respectively, with dupilumab [33,34].

One of the most frequent adverse events in our study was headache that peaked at 16 weeks in eight patients, while no association with an increased risk of skin or systemic infections was demonstrated with continuous long-term dupilumab treatment in adults with moderate-to-severe AD [35].

With these data, the known safety profile of dupilumab is confirmed even in the long term, both in adult patients as well as in adolescents.

The limitation of our study is inherent to its real-life retrospective descriptive nature. Nevertheless, real-life studies report data from daily dermatological practice. Indeed, these studies are generally more generalizable than trials, also including patients who are typically excluded in registration studies, such as patients with multiple comorbidities, and concomitant other treatments [36].

## 5. Conclusions

This real-world study presents results from a large number of patients who were followed from the start of dupilumab therapy for up to three years. The patient characteristics were comparable with those reported in clinical trials and other real-world studies. Overall, EASI, DLQI, and NRS scores decreased significantly during the treatment period from initiation up to three years, demonstrating that treatment efficacy is maintained and improved over time.

The safety profile of dupilumab was also maintained during the three years of treatment, showing no increase in adverse effects with long-term treatment.

These results also confirm the efficacy and safety profiles in continuous long-term treatment in adult and adolescent patients with moderate-to-severe atopic dermatitis.

**Author Contributions:** Conceptualization: M.O., S.R. and L.M.; Methodology: L.M. and N.M.; Software: L.M., S.R. and N.M.; Validation: S.R., P.Q. and M.O.; Formal analysis: L.M. and N.M.; Investigation: M.O., G.C., F.G., N.M. and I.R.; Resources: P.Q. and S.R.; Data Curation: L.M. and N.M.; Writing—Original Draft: N.M.; Writing—Review and Editing: L.M., N.M., P.Q. and S.R.; Visualization: L.M., N.M., P.Q., S.R., M.O., G.C., I.R. and F.G.; Supervision: S.R.; Project administration: S.R. and L.M.; Funding acquisition: P.Q. and S.R. All authors have read and agreed to the published version of the manuscript.

**Funding:** This research received no external funding.

**Institutional Review Board Statement:** This study was approved by our local ethical committee under the protocol SSDERMO13.

**Informed Consent Statement:** Informed consent was obtained from all subjects involved in the study.

**Data Availability Statement:** Data are available upon reasoneble request.

**Conflicts of Interest:** The authors declare no conflict of interest.

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
