# Peer review of "Long-Term Efficacy and Safety of Dupilumab in Patients with Atopic Dermatitis: A Single-Centre Retrospective Study"

_cosmetics, doi:10.3390/cosmetics10060153_

Round 1

Reviewer 1 Report

Comments and Suggestions for Authors

The authors provide real world evidence for the efficacy of dupilumab in treating atopic dermatitis (AD). They prospectively followed a cohort of patients using meaningful clinimetric tools. The following are minor comments.

Figure 1 would benefit from 95% confidence intervals or other display of variability.

What was the source of funding for the study?

The introduction is quite long and it is unclear that readers need such a long description of atopic dermatitis.

The following statement is too strong: "Our study data demonstrated that dupilumab 225 is effective in adult patients..." Not all patients improved, but the majority did improve.

I do not understand the methods statement: "300 mg fl sc every 14 days."

Author Response

Nov. 4th, 2023

Dear Editor,

I am pleased that our paper has been taken into consideration by your honored journal.

I thank the reviewers for the much-appreciated comments which have been extremely helpful in improving our draft.

Below I will give a step-by-step response to the reviewer’s suggestions.

REVIEWER 1

1)  Fig 1: we did an analysis using a chi-square and t-test which do not include confidence intervals

2) There was no source of funding for the study

3) We reduced the introduction, we removed: “The pathogenesis of AD can be divided into the following three main categories- epidermal barrier dysfunction- immune dysregulation- alteration of the microbiome. Each of these can in turn be modulated by genetics and environmental factors” line 50-54 and “Atopic dermatitis, being a chronic relapsing inflammatory skin disease, generally presents in three different clinical phases- acute AD: characterized by an erythematous, intensely itchy papular eruption associated with excoriation and serous exudate10- Subacute AD: with dry, scaly, erythematous papules and plaques- chronic AD: characterized by lichenification due to repeated scratching” line 64-69

4) We change “"Our study data demonstrated that dupilumab 225 is effective in adult patients..." in “In our case series, the majority of patients improved, showing how dupilumab can be a valid treatment option”

5) We change “300 mg fl sc every 14 days” in “300 mg every 14 days”

Reviewer 2 Report

Comments and Suggestions for Authors

Data on IGE levels and LDH levels are missing and are of interest and should be added at least in introduction part.Also data on SCORAD are of interest.

Data on skin barrier measurement like TEWL before and during the treatment are also of interest for the readers.

If available,data on imagistic measurement or the skin parameters like dermoscopy,RCM,OCT are of interest before and durig the treatment so ,if available data from literature should be added in the discussion part.

Data on Funding should be added.

References part should be significally improved-some references has no pages,volumes numbers,some has the complete title of the journal ,not the abbreviation,other the abbreviations,some has month ,other has no months.

This part should be carrefully corrected.reference 12 is unclear 12    1

Some references has PMID from Pubmed.The authors are invited to use the same kind of usage of the references style

Author Response

Nov. 4th, 2023

Dear Editor,

I am pleased that our paper has been taken into consideration by your honored journal.

I thank the reviewers for the much-appreciated comments which have been extremely helpful in improving our draft.

Below I will give a step-by-step response to the reviewer’s suggestions

REVIEWER 2

1) We add in the introduction “The values of immunoglobulin-E, lactodehyde dehydrogenase (LDH) and transepidermal water loss (TEWL) also change during the course of dupilumab therapy, but were not evaluated in our study.” Line 107

2) Data on imagistic measurement or the skin parameters like dermoscopy, RCM, OCT are not available.

3) There was no source of funding for the study

4) We have improved the references

I remain at your disposal for any question or supplementary request.

Best regards,

Luca Mastorino